# From Non-Alcoholic Fatty Liver Disease to Liver Cancer: Microbiota and Inflammation as Key Players

**DOI:** 10.3390/pathogens12070940

**Published:** 2023-07-15

**Authors:** Avilene Rodríguez-Lara, Ascensión Rueda-Robles, María José Sáez-Lara, Julio Plaza-Diaz, Ana I. Álvarez-Mercado

**Affiliations:** 1Center of Biomedical Research, Institute of Nutrition and Food Technology “José Mataix”, University of Granada, Avda. del Conocimiento s/n., Armilla, 18016 Granada, Spain; avilenerl@correo.ugr.es; 2Department of Nutrition and Food Science, Faculty of Pharmacy, University of Granada,18071 Granada, Spain; ruedarobles@ugr.es; 3Department of Biochemistry and Molecular Biology I, School of Sciences, University of Granada, 18071 Granada, Spain; mjsaez@ugr.es; 4Children’s Hospital Eastern Ontario Research Institute, Ottawa, ON K1H 8L1, Canada; 5Instituto de Investigación Biosanitaria ibs.GRANADA, Complejo Hospitalario Universitario de Granada, 18014 Granada, Spain; 6Department of Biochemistry and Molecular Biology II, School of Pharmacy, University of Granada, 18071 Granada, Spain

**Keywords:** non-alcoholic fatty liver disease, cancer, gut–liver axis, hepatocarcinoma, microbiome, inflammation

## Abstract

It is estimated that 25% of the world’s population has non-alcoholic fatty liver disease. This disease can advance to a more severe form, non-alcoholic steatohepatitis (NASH), a disease with a greater probability of progression to cirrhosis and hepatocellular carcinoma (HCC). NASH could be characterized as a necro-inflammatory complication of chronic hepatic steatosis. The combination of factors that lead to NASH and its progression to HCC in the setting of inflammation is not clearly understood. The portal vein is the main route of communication between the intestine and the liver. This allows the transfer of products derived from the intestine to the liver and the hepatic response pathway of bile and antibody secretion to the intestine. The intestinal microbiota performs a fundamental role in the regulation of immune function, but it can undergo changes that alter its functionality. These changes can also contribute to cancer by disrupting the immune system and causing chronic inflammation and immune dysfunction, both of which are implicated in cancer development. In this article, we address the link between inflammation, microbiota and HCC. We also review the different in vitro models, as well as recent clinical trials addressing liver cancer and microbiota.

## 1. From Non-Alcoholic Fatty Liver Disease to Liver Cancer

Twenty-five percentage of the world’s population suffers from non-alcoholic fatty liver disease (NAFLD) [1]. The main signature of this disease is an increment in fat accumulation, in the form of micro and macro vacuoles of lipids into hepatocytes (>5% fat content in the liver; referred to as steatosis) [2].

Between 10 and 25% of patients with NAFLD may progress to non-alcoholic steatohepatitis (NASH) [3]. NASH’s main features are hepatic steatosis in association with inflammation and ballooning with progressive collagen deposition and subsequent vascular remodeling [4]. Its progression deeply aggravates the risks of cirrhosis, liver failure and hepatocellular carcinoma (HCC) [3,5,6,7].

The liver is constantly exposed to metabolites, toxins, and microbial products from the intestine, due to its blood supply through the portal vein. It is equipped with several immune mechanisms to prevent an excessive inflammatory response when it is exposed to a normal antigen load. These mechanisms include downregulation of the expression of key histocompatibility system proteins, suppression of antigen presentation by Kupffer cells and dendritic cells and upregulation of immunosuppressive cells such as regulatory T cells. However, these mechanisms are overwhelmed in the setting of NAFLD because increased fat accumulation in the liver leads to cellular damage, mitochondrial dysfunction, oxidative stress and activation of cell death pathways. Altogether, these trigger chronic liver inflammation that leads to the progression of NASH but also contributes to the development of HCC [8].

Despite numerous advances, the pathophysiology of NAFLD has not yet been fully described. The combination of factors that lead to NASH and its progression to HCC against the background of inflammation is also not clearly understood. Although evidence supports the effects of human microbes on cancer development and their contribution from different dimensions, the results of preclinical studies are difficult to translate to the clinic.

In this paper, we explore the link between inflammation, microbiota and HCC. We also review the different in vitro models, as well as recent clinical trials addressing liver cancer and microbiota.

## 2. The Bidirectional Liver–Gut Communication and Its Impact on Liver Disease

The portal vein is the main connection channel between the intestine and the liver, allowing the direct transport of products from the intestine to the liver. In addition, this connection also permits feedback from the liver to the intestine through the secretion of bile and antibodies [9]. This relationship is known as the gut–liver axis (Figure 1).

One of the determining factors in the gut–liver axis relationship is intestinal permeability, which refers to the ability of the intestinal lining to allow substances to pass through it. The intestinal lining is made up of cells held together by tight junctions that act as a barrier to prevent unwanted substances from passing into the bloodstream, known as a leaky gut [10]. A leaky gut can be a gateway for toxic substances, which can limit the liver’s ability to purify, filter and cleanse the system.

Although diet, microbiota, and gut mucosa are independent, they are interconnected. They are also all connected to the host through the gut–liver axis. Intestinal products regulate bile acid synthesis, hepatic glucose, and lipid metabolism [11] and as well as microbiota composition and intestinal barrier function [12]. For instance, choline presents important effects on hepatic lesions such as its involvement in the metabolism of fats in the liver. Dietary choline has been associated with HCC mortality [13]. However, findings from mouse models indicate that a diet deficient in methionine and choline induces severe hepatic steatosis and inflammation [14].

Also, short-chain fatty acids (SCFAs), such as acetate, propionate, and butyrate, contribute to maintaining gut health and regulate various physiological processes. They can also have indirect effects on the liver and hepatic lesions through their interactions with the gut–liver axis. However, it is important to note that the specific effects of SCFAs on hepatic lesions are still an active area of research. On this note, a recent systematic review assessing the impact of SCFA supplementation on liver injury and intestinal permeability indicated that SCFA supplementation in liver disease ameliorates liver injury by maintaining gut epithelial integrity [15].

The impact of bile acids on hepatic lesions can be significant and multifaceted, since, in addition to regulating bile flow, lipid metabolism, and immunity, they are primarily synthesized and metabolized in the liver. Moreover, bile acids can interact with gut microbiota, which further influences their effects on hepatic lesions. In this sense, both cirrhotic and non-cirrhotic patients with NASH-HCC have shown a clear association between altered gut microbiota and primary conjugated bile acid composition [16]. Some authors have even suggested that the decreasing percentages of conjugated deoxycholic acids in serum may be closely related to HCC, which can be induced by gut bacteria. Another study performed by Thomas et al. concluded that primary conjugated bile acids are more strongly associated with an increased risk of HCC, whereas secondary over primary bile acid ratios are significantly associated with a lower risk. These authors further proposed that modifying the gut microbiota to modulate bile acid metabolism could serve as a viable approach for the primary prevention of HCC in individuals with metabolic dysfunction and fatty liver disease [17].

A leaky gut facilitates the entry of pathogen-associated molecular patterns (PAMPs), such as lipopolysaccharide (LPS or endotoxin), and microbiome-derived metabolites into the liver. This triggers a proinflammatory response that aggravates liver inflammation. In this context, the intestinal bacterial load determines the amount of PAMPs entering the portal and systemic circulation, which increases the severity of liver inflammation. The progression of chronic liver disease from a compensated to a decompensated phase is associated with impaired intestinal defense mechanisms, resulting in further impairment of intestinal barrier function [18].

The most significant impairment of the intestinal barrier is seen in advanced end-stage liver disease, specifically in decompensated cirrhosis. Functional effects resulting from altered communication between the gut and liver are also found in several chronic liver diseases, in which the liver’s innate immune cells are repeatedly exposed to bacterial products from the gut, such as endotoxins, and metabolites such as ethanol and trimethylamine. These exposures lead to liver inflammation [4].

Accumulating evidence suggests that disruption of the gut–liver axis contributes to chronic liver diseases, including cirrhosis [11].

The main signs of changes in the gut–liver axis that led to NAFLD include altered gut microbiota, altered intestinal barrier, and thus increased permeability, and altered luminal bile acid levels. In turn, altered bile acid levels reduce intestinal FXR signaling, compromising intestinal mucosal and antimicrobial peptide synthesis and intestinal mucosal and intestinal–vascular barrier integrity [19]. The relative contribution of each of these abnormalities to the disruption of the intestinal heme–intestinal axis depends on the etiology and stage of liver disease [20,21].

Several factors can alter the functional connection between the gut and liver, ranging from diet, genetics, and environment [20]. Excessive alcohol consumption impairs the integrity of the tight junctions of the intestinal barrier. This results in increased permeability, inflammation of the intestine, and modification of the intestinal microbiota composition. A diet rich in fats and sugars may also contribute to fat accumulation in the liver [22].

In addition, some foods are detrimental to the gut–liver axis, such as excess salt in the diet, which can result in increased blood pressure and fatty liver disease. Refined sugars present in foods such as pastries, soft drinks, and sweets, among others, can increase blood sugar levels and contribute to liver fat accumulation [23]. On the contrary, some foods can support optimal interaction between the gut and the liver, such as foods rich in vitamins C and E, omega-3, carotenes, and some flavonoids, because they have antioxidant, anti-inflammatory, and liver-protective properties (artichoke, salmon, garlic, broccoli) [24].

The microbiota’s composition undergoes changes that alter its functionality (dysbiosis). A state of inflammation may arise that alters the intestinal barrier. This may result in a loss of integrity and the passage of microorganisms and metabolites [25]. Genetic factors also affect the interaction between the gut and liver axis. Some inherited liver diseases such as hemochromatosis, Wilson’s disease, and alpha-1 antitrypsin deficiency affect liver function. It has also been suggested that genetic factors may lead to NAFLD [26].

## 3. The Link between Inflammation, Microbiota and Hepatocarcinoma

Generally speaking, cancer is a medical condition that includes several diseases characterized by the uncontrolled growth of cells in the body. This unregulated proliferation leads to abnormal cells that infiltrate surrounding tissues and organs, resulting in a wide range of adverse health outcomes [27]. Cancer can be caused by genetic mutations, exposure to carcinogenic substances, and lifestyle choices. Due to cancer’s potential severity and complexity, effective prevention and treatment strategies are critical to improving health outcomes for those affected by this condition [28].

Inflammation is a complex physiological process triggered by various stimuli, including injury, infection, or irritation [29]. This process involves the activation of the immune system, the release of a variety of chemical mediators, and the recruitment of immune cells to the affected tissue or organ. The ensuing immune response neutralizes pathogens, clears debris, and facilitates tissue repair [27,30]. However, excessive or prolonged inflammation can lead to long-term tissue damage, impaired organ function, and various pathological conditions. It is well-established that inflammation is a key factor in tumor predisposition and promotion [27,30]. In this sense, several studies have found a link between cancer and inflammation. Chronic inflammation is suggested as a contributing factor to cancer development and progression [27,30] that can induce the release of reactive oxygen and nitrogen species, cytokines, and chemokines. These can damage DNA and promote cancer cell proliferation [31].

There are also many proinflammatory and inflammatory factors released by leukocytes and mast cells that induce the development of cancer. These factors include interleukin (IL)-6, IL-1, nuclear factor kappa B (NF-κB), tumor necrosis factor, (TNF)-alpha, signal transducer and activator of transcription 3 (STAT3), and transforming growth factor-β (TGF-β), among others, and they play a significant role in chronic inflammation [32]. Moreover, chronic inflammation suppresses the immune system, creating a microenvironment that is conducive to tumorigenesis, tumor development, and metastatic growth [33]. Several authors have described that chronic inflammation can increase the risk of developing liver cancer [34,35]. In fact, liver cancer is associated with chronic hepatitis [36]. The liver microenvironment plays a crucial role in the pathogenesis of HCC, as chronic inflammation, driven by factors inside the liver, facilitates the progression of cirrhosis and HCC. Hepatic stellate cells and tumor macrophages contribute to the induction of fibrosis through the production of the extracellular matrix and promote tumor growth, leading to the process of angiogenesis deeply linked to hepatic inflammation [37]. Inflammatory cells such as neutrophils, macrophages and lymphocytes can also produce growth factors and enzymes that support tumor growth and invasion [38].

Neutrophil extracellular traps (NETs) are implicated in the pathogenesis of NASH. Their association with inflammation and globular degeneration highlights their role in the disease. Some studies suggest that the fibrous structure of NETs enhances their bactericidal capacity by sequestering bacteria with a high local concentration of antimicrobial molecules [39]. In addition, IL-1β- and IL-17A-enriched NETs contribute to the hepatic inflammatory process in NASH by providing a vehicle for IL-1β and IL-17A. Furthermore, platelet aggregation in hepatic sinusoids implicates the role of thrombo-inflammation in NASH and may explain the low peripheral blood platelet counts observed in these patients [40].

The two main pro-tumorigenic mechanisms by which immune cells promote HCC include the secretion of cytokines and secretion of cytokines and growth factors that promote proliferation or counteract apoptosis of tumor cells, as well as suppress the anti-tumor function of lymphocytes. In addition, reported results indicate that the NF-κB and JAK-STAT pathways are key inflammatory signaling pathways in the promotion of HCC [41].

The role of gastrointestinal B cells in the development of NASH, fibrosis and NASH-induced HCC was also examined. Activated B cells in the gut were elevated in both human and mouse NASH samples. These activated B cells were shown to contribute to NASH development independently of antigen specificity and gut microbiota by promoting the metabolic activation of T cells [42].

In addition, viruses and bacteria may also cause chronic inflammation, which contributes to cancer. In this regard, a study by Simon et al. performed in adults with histologically defined NAFLD in Sweden from 1966 to 2016 revealed a higher incidence of HCC in patients with biopsy-proven NAFLD compared to the controls [43]. In addition, the presence of NASH increases the risk of developing HCC compared to patients without this condition [44].

The association between NAFLD and cancer may also be mediated by other metabolic traits, such as obesity or diabetes. In this regard, the study by Kanwal et al. in a cohort of 271,906 patients diagnosed with NAFLD indicated that each additional metabolic trait elevates the risk of cirrhosis and HCC in patients with NAFLD. The presence of diabetes was the most strongly associated with HCC in the presence or absence of cirrhosis [45].

The study conducted by Yang et al. on patients with NASH also highlights the association between diabetes and increased HCC risk [46].

Along the same line, insulin resistance and obesity are also linked to chronic inflammation and an increased risk of HCC [47]. Insulin resistance is a condition where the body’s cells become resistant to insulin effects, leading to high blood glucose levels. This can induce oxidative stress, which is an imbalance between reactive oxygen species production and the body’s antioxidant defenses [48]. Furthermore, obesity is a factor associated with chronic inflammation and liver damage [49,50], which in turn can also lead to insulin resistance and cancer. Insulin boosts insulin-like growth factor 1 (IGF-1) production in the liver by increasing growth hormone receptors. When IGF-1 binds to its receptor, IGF-1R, it promotes cancer cell growth. Additionally, IGF-1 triggers anti-apoptotic activity through various signaling systems, including the phosphoinositide 3-kinase (PI3K)/Akt signaling pathway and the mitogen-activated protein kinase (MAPK) pathway [51,52].

### 3.1. Dysbiosis and Liver Cancer

A healthy gut is typically inhabited by diverse collections of bacteria and other microbes. Dysbiosis refers to a disruption in this balance, in which the normal bacterial content, metabolic functions, or distribution within the gut is altered. This phenomenon can be associated with various diseases [53]. This term is used to describe the imbalance in microbiota. Dysbiosis has been associated with multiple diseases, including cancer. Various bacterial species are involved in the initiation and advancement of human cancers through diverse mechanisms. These bacterial species commonly employ several strategies such as triggering inflammation, modifying cell signaling, promoting invasion and immune evasion, colonizing specific niches, inducing DNA damage and mutations, expressing specific microRNAs, and amplifying epigenetic effects. These mechanisms collectively contribute to alterations in the cell cycle [54,55,56,57].

The relationship between dysbiosis and HCC is a topic of increasing interest in the scientific community. Research has suggested that dysbiosis may play a role in the development and progression of HCC, particularly in individuals with liver cirrhosis.

This result is in line with a study conducted by Zhang et al. on patients with primary liver cancer and liver cirrhosis. These authors found a trend toward an increase in *Enterobacter ludwigii* species when compared to the control and cirrhosis groups. As the disease progressed, the *Firmicutes* to *Bacteroidetes* ratio decreased substantially [58]. The study also showed that the genera *Streptococcus*, *Prevotella*, *Actinomyces*, *Veillonella*, and *Neisseria* were the dominant genera observed in liver cancer patients’ saliva, suggesting a potential correlation between oral microbial imbalances and liver cancer incidence [59].

Significant changes in gut microbiota have been observed in early HCC patients. Ren et al. conducted a study addressed to evaluate fecal samples from individuals with initial HCC, cirrhosis, and healthy controls. These authors found a notable increment in *Actinobacteria*, *Gemmiger*, and *Parabacteroides* species in early HCC patients compared to cirrhotic individuals. The study also showed a reduction in the abundance of butyrate-producing bacterial genera and an increase in LPS-producing genera in patients with early HCC compared to the controls [60]. Similar findings were previously reported by Romana-Ponziani et al. [61], who investigated the microbiota characteristics associated with HCC in patients with NAFLD-related liver cirrhosis. They observed a higher abundance of *Enterobacteriaceae* and *Streptococcus*, and a decrease in *Akkermansia* in cirrhotic patients versus healthy controls. Furthermore, the study revealed a microbiota enriched with *Bacteroides*, *Ruminococcaceae*, *Enterococcus*, *Phascolarctobacterium*, and *Oscillospira* in cirrhotic individuals, plus a first-time diagnosis of HCC. These findings are in line with Li et al. They observed lower levels of *Akkermansia* in both NASH-HCC patients and mice [62]. Similarly, Montaresser et al. conducted a study [63] aimed to assess the potential impact of imbalances in the gut microbiota on the development of HCC in individuals with chronic hepatitis C virus infection. These authors reported that there were no significant differences between the HCC patients’ group and the healthy group regarding the presence of *Bacteroides fragilis* and *Akkermansia muciniphila*. However, they found that *Faecalibacterium prausnitzii* and *Bifidobacterium* were less prevalent in HCC patients (51% and 43%, respectively) than in healthy controls. Moreover, *Lactobacillus* and *Escherichia coli* were more commonly found in HCC patients than in healthy controls. Zheng et al. also investigated the impact of gut dysbiosis on liver cirrhosis-induced and non-induced HCC (LC-HCC and NLC-HCC, respectively) by analyzing fecal samples from individuals with hepatitis, cirrhosis, and HCC. They found that gut microbial diversity was closely associated with the presence or absence of liver cirrhosis in patients with HCC, rather than HCC itself. Additionally, the groups with liver cirrhosis had significantly higher levels of the *Fusobacteria* and *Proteobacteria*, and lower levels of the Tenericutes phylum compared to the LC-HCC and NLC-HCC groups [64]. Accordingly, Effenberger et al. compared profiles to non-malignant cirrhotic and non-cirrhotic NAFLD patients and found that patients with HCC and cirrhosis exhibited an increased presence of bacterial gene signatures in contrast to NAFLD [65].

As mentioned, dysbiosis can produce persistent inflammation and impaired immune function, which have both been linked to the development of cancer. [66]. In this regard, the study by Behary et al. characterized the gut microbiota of NAFLD patients using metagenomic and metabolomic analysis. Their results suggest that the gut microbiota in NAFLD-HCC patients presents a distinctive microbiome/metabolomic profile, and may modulate the peripheral inflammatory response [67]. Similarly, a cross-sectional study conducted by Zhang et al. examined the dysbiotic profile, microbial translocation, and intestinal damage at various stages of HCC [68]. Their results revealed a significant decrease in the abundance of the *Bifidobacteriaceae* family during the initial, intermediate, and terminal stages of HCC, while the abundance of the *Enterococcaceae* family increased significantly. Additionally, HCC progression was associated with an elevation in inflammatory cytokine levels, accompanied by an immunosuppressive T-cell response and microbial translocation [68].

Even though there were no significant differences in terms of alpha diversity, the main component analysis of the Bray–Curtis distance revealed a significant clustering of fecal microbiota between HCC patients and healthy volunteers [68].

### 3.2. Microbiota Manipulation in Clinical Trials for Patients with Liver Cancer

Based on the information available on the clinicaltrial.gov website and the keywords “liver cancer” and “microbiota,” 17 studies were displayed. Of the seventeen studies, three have been completed with no information regarding publications, one has been withdrawn, three have an unknown status, two are not yet recruiting, two have been invited to enroll, one is active and not recruiting, and five are recruiting. We summarize the clinical trials related to liver cancer and microbiota manipulation in Table 1.

Only one clinical trial has been completed from those 17 registered studies. As part of this project, a bifidobacteria-rich product will be used as an intervention drug to sustain medication for patients with HCC undergoing hepatectomy perioperatively, and recovery of liver function will be observed postoperatively.

As a result of immunotherapy, significant bacterial differences were observed between patients with objective tumor responses and those with progressive disease (by Anosim and Adonis tests). HCC patients with progressive disease to immunotherapy with checkpoint inhibitors showed a higher prevalence of *Prevotella*, usually considered a pathogenic bacterium. In patients with objective tumor responses, *Lachnospiraceae, Veillonella, Lactobacillales, Lachnoclostridium, Streptococcaceae* and *Ruminococcaceae* predominate. In addition, primary bile acids, including α and β-muricholic acids, and murocholic acid, as well as secondary bile acids, such as ursocholic acid, tauro-ursodeoxycholic acid, ursodeoxycholic acid, and taurohyocholic acid, were significantly predominant in the patient’s feces with objective tumor responses to immunotherapy with checkpoint inhibitors treatment. *Lachnoclostridium, Ruminococcus*, and secondary bile acids were significantly related to correlation networks in patients with objective tumor responses [69].

Nivolumab 3 mg/kg plus ipilimumab 1 mg/kg was administered to patients with potentially resectable HCC once every three weeks (N+I). Following two to four cycles of N+I, the recruited subjects were screened for surgery. For 16S rRNA sequencing, samples of stool were collected before, after, and at the end of the first cycle of N+I treatment to extract bacterial DNA. In terms of the diversity of the gut microbiota, there was no significant difference between the subjects with and without tumor progression, but there was a significant difference in the abundance of some bacteria in the subjects without tumor progression [70].

Based on Mendelian randomization, *Ruminococcaceae* and *Porphyromonadaceae* were related to HCC, and *Porphyromonadaceae* and *Bacteroidetes* were associated with intrahepatic cholangiocarcinoma. Based on the case–control study, the authors validated their findings with sequencing data. After statistical analysis, the relative abundance of relevant gut microorganisms was higher in healthy controls than in patients, suggesting a causal link. According to this study, *Ruminococcaceae, Porphyromonadaceae*, and *Bacteroidetes* may be associated with a decreased risk of liver cancer (HCC or intrahepatic cholangiocarcinoma), suggesting a potential role in preventing and controlling this disease [71].

It is important to note that in other diseases, manipulation of microbiota has a greater impact than in HCC. It has been demonstrated that animal models of NASH respond to fecal microbiota transplantation (FMT), and early studies have shown that FMT from lean mice donors results in changes in the gut microbiota of obese mice, which are thought to be primarily related to increased microbiota diversity. An ongoing phase I clinical study (NCT02469272) examining small intestinal microbiota transfers from lean to obese subjects has shown improved insulin sensitivity in patients with metabolic syndrome as well as improved insulin sensitivity in those with NASH [37,72,73]. Several factors, including the strength and adaptability of host and donor characteristics, contribute to the variable efficacy of FMT. Research in this area is promising and requires high-quality studies and controlled trials in NASH patients [73,74].

## 4. In Vitro Models of Liver Cancer

To establish the final relationship between microbiota and liver cancer, further research and models of study are necessary. Physiological and pathological aspects of liver diseases have gained greater understanding over the years as a result of research. Overall, rodent models were unable to accurately predict less than 50% of the therapeutic response and toxicity of drugs that are clinically used in humans [75]. To study biological aspects of tumors [76,77], pharmacological mechanisms, efficacy, and toxicity, in vitro human cell cultures are the preferred model to rodent models [75].

Historically, in vitro studies have been conducted using bi-dimensional cell lines derived from hepatocarcinoma and hepatoma, as well as primary cultures of the cells, providing a useful tool to study and characterize molecular events at the base of disease onset and progression, and to obtain information regarding treatment efficacy [78]. One of the most commonly used preclinical experimental models for HCC research is HepG2, a cell line derived from a liver biopsy of a Caucasian adolescent [76]. HepG2 displays the characteristic features of a hepatic lesion, such as the increased expression of α-fetoprotein, and expresses distinct functions of hepatic cells, including glycogen synthesis, plasmatic protein synthesis, biliary acid synthesis, and cholesterol and triglyceride metabolism [76].

Most preclinical research on cholangiocarcinoma has been conducted using human extrahepatic cholangiocarcinoma cells, TFK-1 (an extrahepatic bile duct carcinoma specimen collected from a 63-year-old man was described in the literature as expressing the c-erbB-2 protein) and EGI-1 (in 1984, it was established from a solid tumor from a 52-year-old man with advanced stage malignant bile duct carcinoma (the patient had not been treated with chemotherapy before cell transplantation) and three consecutive passages in female nu/nu mice; primary tumor histology: adenocarcinoma of low differentiation, metastasizing (ascites inpatient); human karyotype confirmed by passage #34 of in vitro culture (1986)), and intrahepatic cholangiocarcinoma cells, HuCC-T1 (in 1989, it was established from a bile duct carcinoma from a 56-year-old patient, derived from metastasis ascites) and RBE (adenocarcinoma cholangiocarcinoma, from a female Japanese patient), derived from malignant ascites [77]. These cells are representative of a single subtype of cholangiocarcinoma and do not offer sufficient data to comprehensively examine its molecular biology [79].

A primary hepatocyte’s lifespan in culture is limited, lasting only a few days, which leads to a decline in hepatic function in vitro [80] and necessitates the donation of fresh material, which is expensive [78]. In addition, primary cultures must be derived with great care since it is possible to detect an unwelcome increase in healthy cell fractions that must be removed [77]. While 2D cell techniques have several advantages, such as easy reproducibility and lower costs, tumor tissue remains too heterogeneous and characterized by complex and dynamic microenvironments, despite multiple advantages [81].

Researchers have been working in recent years to develop three-dimensional (3D) cell models that may be generated from both biopsies and commercially available 2D cell lines. We have summarized four 3D models, namely spheroids, scaffold-based systems, bioprinting models, and organoids.

### 4.1. Spheroids

A 3D system represented by a spheroid is one of the first known 3D systems, which exhibit a cylindrical shape and are enriched with stem-like cells; however, this system is too simple to mimic the organization of tumors [81]. A single-cell suspension or a multi-cell suspension of primary cultures can be used to produce spheroids [82]. Floating spheres can typically be developed when the single-cell suspension is maintained without a matrix, on ultra-low attachment plates, and without serum [83]. Besides drug screening and immune interaction modeling [84], spheroid systems may also be used for establishing co-culture systems containing both healthy and cancerous cells so that angiogenesis and tumor metastatic mechanisms can be studied [85].

### 4.2. Scaffold-Based System

A scaffold-based system allows cells to be embedded into a physical matrix, enabling them to aggregate, proliferate, and migrate [76]. To simulate the microenvironment of tissues and tumors, scaffolds consist of a variety of materials that vary in porosity, permeability, and mechanical stability [86]. Hydrogels are one of many types of scaffolds that can mimic the characteristics of the extracellular matrix, allowing soluble factors, such as cytokines and growth factors, to pass through the gel support like it was a tissue [87]. Considering their adaptability, hydrogels can be prepared to meet the needs of various experiments. In addition to natural hydrogels, synthetic hydrogels are made with polymeric materials with chemically defined bases, such as polyethylene glycol and polylactate, typically made of natural polymers such as fibrinogen, collagen, hyaluronic acid, gelatin, and alginates [88]. Matrigel is one of the most widely used natural hydrogels in 3D cell culture. In this substance, collagen IV, laminin, proteoglycans, soluble heparan, and entactin are found in abundance and can solidify at 37 °C, mimicking the properties of the base membrane matrix derived from Engelbreth–Holm–Swarm tumors [89].

### 4.3. Bioprinting

Bioinks used in 3D bioprinting enable the creation of 3D constructs with tissue-like architecture using living cells, decellularized extracellular matrix constituents, nutrients, growth factors, and biomaterials [90,91]. Therefore, bioprinting technology is capable of reproducing the extracellular matrix successfully, which may enhance cellular proliferation rates and the response to chemotherapeutic drugs when compared to traditional 2D models [92]. In addition, organ-on-a-chip models mimic real and synthetic microenvironments, integrating living cells capable of emulating an organ’s function in vitro. Several organ-on-a-chip models have been combined in new studies to create body-on-a-chip models that represent multi-organ interactions and allow a better understanding of the metastasis process in cancer [81].

### 4.4. Organoids

Essentially, organoids are in vitro 3D models resembling some of the structures and functions of their in vivo counterparts, which cannot be seen in 2D cultures. They are created by separating specialized epithelial tissues, using embryonic stem cells or induced pluripotent stem cells, which are capable of self-renewal and self-organization [85].

It is necessary to identify mitogenic signals in human liver organoids derived from adult tissues. These signals include fibroblast growth factor, epithelial growth factor, and hepatocyte growth factor [93,94,95]. It has been demonstrated that forskolin is an activator of cyclic adenosine monophosphate and A8301 inhibits transforming growth factor-β signaling, allowing patients to expand on such a medium for an indefinite period [93,94]. To prevent apoptosis, ROCKi, an inhibitor of Rho-associated kinase protein, is added to the medium a few days after seeding [93,96].

Organoids have the potential to open the door to the regeneration of injured or diseased organs in the future, a proposition thought to be unlikely in medicine up until now. Due to the potential for liver organoids to regenerate diseased livers, current research is focused on creating organoid liver buds for delivery via the portal vein to patients who need liver transplants urgently [97]. The future of personalized medicine may thus require all patients to maintain organoid tissue in large-scale biobanks to implement a structured system of patient-centered treatment. A closer collaboration with bioengineers is also important, and adding blood vessels to liver organoids would be a reasonable approach to overcoming the problem of limited nutrition availability that ultimately affects the growth of organoids [98].

### 4.5. Microbiota-Based Models of Liver Cancer In Vitro

There are several ways in which single strains of microbiota or mixtures of several strains may be added to the hepatic cell line cultures, as shown in the following examples. The polysaccharide *Pleurotus ostreatus* significantly reduced tumor cell metastasis. The partially pure polysaccharide treatment resulted in a decrease in Foxp3 and Stat3 expression and an increase in immunological factors, such as IL-2, tumor necrosis TNF-alpha, and interferon-gamma [99]. A significant increase in peroxisome proliferator-activated receptors (PPAR)-gamma and PPAR-alpha expression is induced by *Bifidobacterium longum* when compared with basal conditions. It is interesting to note that this anti-inflammatory effect was also observed in HepG2 cells when they were stimulated with LPS [100].

A limited amount of information is available regarding liver cancer in terms of 3D models and microbiota. By using liver organoids and 3D bioprinting technology, we will be able to model multiple processes of the disease, interactions with different cell types, and individual patient heterogeneity in an attempt to better simulate the relationship between the microbe and liver disease. In Figure 2, we summarize the main in vitro models used in the study of liver cancer.

## 5. Further Perspectives

Even though NAFLD has become a widespread disease, its exact cause is unknown and varies from patient to patient. NAFLD is linked to obesity, and metabolic syndrome and its associated features including abdominal obesity, insulin resistance, glucose intolerance or type 2 diabetes mellitus and atherogenic dyslipidemia [101,102].

Not all patients with these conditions progress to the development of HCC. The reason why some patients only develop steatosis and others develop NASH and liver cancer is not completely elucidated. This is probably the result of numerous metabolic anomalies in the context of a genetic predisposition [103]. In this concern, the relationship between HCC and inflammation, as well as the metabolic interaction between the gut and liver in both homeostasis and dysbiosis, is complex and multifaceted. Dysbiosis can alter the gut–liver axis and promote liver inflammation and damage, leading to HCC. Unfortunately, there are many players on the court whose individual weight, as well as the combination of several or all of them, can break the balance that allows the patient with steatosis to progress to a more serious state. Thus, more research is needed to fully understand the mechanisms underlying this association.

Even advances in knowledge about the gut–liver axis would contribute to the development of microbiota-based diagnostic, prognostic, and therapeutic tools for HCC. The identification of a panel of microbes that will be used as signs of liver damage and disease progression would contribute to predicting hospitalization, bacterial infection, and complications.

Therefore, further investigation is needed to better understand the mechanisms underlying the relationship between dysbiosis and HCC and identify potential strategies for manipulating the gut microbiome to prevent and treat liver cancer.

## Figures and Tables

**Figure 1 pathogens-12-00940-f001:**
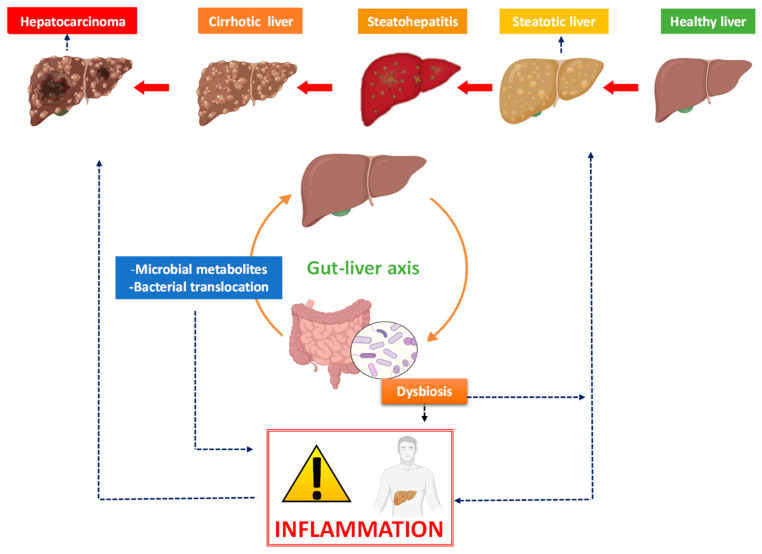
The bidirectional liver–gut communication and its impact on liver disease. In the complex interaction between the intestine and the liver, the portal vein is the main route of communication between the two organs. This connection also allows feedback from the liver to the gut through bile and antibody secretion, transport of microbial metabolites, as well as bacterial translocation. The latter two processes can induce inflammation that contributes to the progression of non-alcoholic liver disease.

**Figure 2 pathogens-12-00940-f002:**
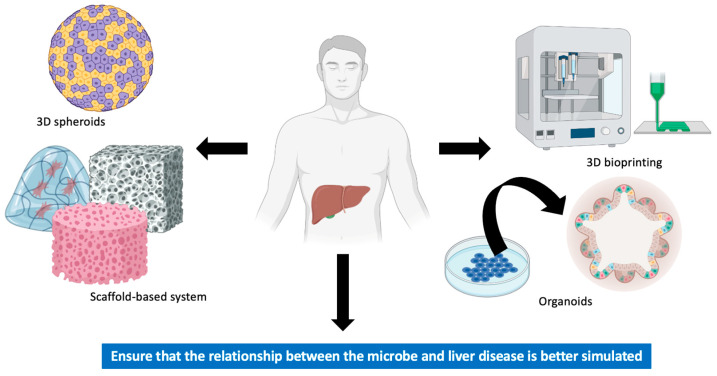
Main in vitro models used in the study of liver cancer.

**Table 1 pathogens-12-00940-t001:** Clinical trials related to liver cancer and microbiota manipulation.

Study Status	Study Title	Study Type	Locations
Recruiting	Anesthesia on gut microbiota and metabolomics, NCT04767503	Interventional	Taiwan
Withdrawn	Gut microbiota in people with HCC, NCT02599909	Observational	
Unknown	Microbiota study in liver transplanted patients, NCT03507140	Observational	France
Enrolling by invitation	Relationship between microbiota and prognosis of HCC after systemic treatments, NCT05443217	Observational	China
Not recruiting	FMT in liver cancer to overcome resistance to atezolizumab and bevacizumab (flora), NCT05690048	Interventional	Germany
Completed	The effect of gut microbiota on postoperative liver function recovery in patients with HCC, NCT04303286	Observational	China
Recruiting	Prebiotic effect of eicosapentaenoic acid treatment for colorectal cancer liver metastases, NCT04682665	Observational	United Kingdom
Completed	Clinical study on Bifico accelerating postoperative liver function recovery in patients with HCC, NCT05178524	Interventional	China
Recruiting	Early detection of HCC in a high-risk prospective cohort, NCT04965259	Observational	Singapore
Recruiting	A multicentre study on features of the gut microbiota of patients with critical chronic diseases, NCT05638269	Observational	China
Unknown	Probiotics in the prevention of HCC in cirrhosis, NCT03853928	Interventional	
Enrolling by invitation	Volatiles in breath and headspace analysis, diagnostic markers, NCT03228095	Observational	Latvia
Not recruiting	FMT in refractory HCC, NCT05750030	Interventional	Austria
Completed	Tumor microenvironment surveillance on simultaneous liver metastases extensive stage small cell lung cancer, NCT05055999	Observational	China
Recruiting	A prospective cohort study of changes in circulatory microRNA of resected HCC, NCT05148572	Observational	Singapore
Unknown	HCC in patients with cirrhosis due to alcohol or a NAFLD, NCT03307408	Observational	France
Active not recruiting	Guangzhou nutrition and health study (GNHS), NCT03179657	Observational	China

Abbreviations: FMT, fecal microbiota transplantation; HCC, hepatocellular carcinoma; NAFLD, non-alcoholic fatty liver disease.

## Data Availability

Not applicable.

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
