# Peer review of "From Non-Alcoholic Fatty Liver Disease to Liver Cancer: Microbiota and Inflammation as Key Players"

_pathogens, 2023, doi:10.3390/pathogens12070940_

Round 1

Reviewer 1 Report

This review aims at the relationship between cirrhosis, hepatocellular carcinoma and gut microbiota, emphasizes the promotion effect of chronic inflammation on HCC progression, and summarizes the application of 3D cell models in drug screening. In general, the review provides a comprehensive information and new option for researchers.

 However, there are a few minor concerns, as listed below:

1.       As PAMPs, LPS generally accelerates HCC development, but bile acids have a disparate impact on disease progression. To add some contents related to the effects of bile acids or other PAMPs such as SCFAs and choline on hepatic lesion will be interesting and valuable.

2.       Analysis of gut microbiota in the clinical study is descriptive, whether some solid evidences from animal models or NASH patients could be cited to ensure bacterial function in some diseases?

3.       It is generally recognized that chronic inflammation facilitates cirrhosis or HCC progression, if possible, the immune microenvironment could be further discussed in more details in a revised version.

4.       Remove “non” from line 160 page 4 where it says “non-NASH increases the risk of developing HCC compared to patients without NASH”.

Author Response

Ms. Nadja Spasojevic,

Assistant Editor, MDPI Belgrade,

Thank you for providing us with the opportunity to submit a revised version of our manuscript entitled “From nonalcoholic fatty liver disease to liver cancer: microbiota and inflammation as key players” to the Pathogens journal. We would like to thank the reviewers for their thoughtful comments and suggestions regarding our manuscript. All comments have been considered and incorporated into the revised manuscript as well as the changes according to the Authenticate report. Changes are highlighted in green and blue font for the reviewer's comments with an itemized point-by-point response to the reviewers' comments.

COMMENTS FROM REVIEWER #1

This review aims at the relationship between cirrhosis, hepatocellular carcinoma and gut microbiota, emphasizes the promotion effect of chronic inflammation on HCC progression, and summarizes the application of 3D cell models in drug screening. In general, the review provides comprehensive information and new option for researchers.

 However, there are a few minor concerns, as listed below:

  1. As PAMPs, LPS generally accelerates HCC development, but bile acids have a disparate impact on disease progression. To add some contents related to the effects of bile acids or other PAMPs such as SCFAs and choline on hepatic lesion will be interesting and valuable.

Response:  According to the reviewer’s comment new information was added and now states (pages 2-3, lines 80-120),

“For instance, choline presents important effects on hepatic lesions such as its involvement in the metabolism of fats in the liver.  Dietary choline has been associated with HCC mortality [13]. However, findings from mouse models indicate that a diet deficient in methionine and choline induces severe hepatic steatosis and inflammation [14]. Also, short-chain fatty acids (SCFAs), such as acetate, propionate, and butyrate contribute to maintaining gut health and regulate various physiological processes. They can also have indirect effects on the liver and hepatic lesions through their interactions with the gut-liver axis. However, it is important to note, that the specific effects of SCFAs on hepatic lesions are still an active area of research. In this concern, a recent systematic review assessing the impact of SCFAs supplementation on liver injury and intestinal permeability indicated that SCFAs supplementation in liver disease ameliorates liver injury by maintaining gut epithelial integrity [15]. The impact of bile acids on hepatic lesions can be significant and multifaceted, since, in addition to regulating bile flow, lipid metabolism, and immunity, they are primarily synthesized and metabolized in the liver. Moreover, bile acids can interact with gut microbiota, which further influences their effects on hepatic lesions. In this sense, both cirrhotic and non-cirrhotic patients with NASH-HCC have shown a clear association between altered gut microbiota and primary conjugated bile acid composition [16]. Some authors have even suggested that the decreasing percentages of conjugated deoxycholic acids in serum may be closely related to HCC, which can be induced by gut bacteria. Another study performed by Thomas et al. concluded that primary conjugated bile acids are more strongly associated with an increased risk of HCC, whereas secondary over primary bile acid ratios are significantly associated with a lower risk. These authors further proposed that modifying the gut microbiota to modulate bile acid metabolism could serve as a viable approach for the primary prevention of HCC in individuals with metabolic dysfunction and fatty liver disease [17]”.

  1. Analysis of gut microbiota in the clinical study is descriptive, whether some solid evidence from animal models or NASH patients could be cited to ensure bacterial function in some diseases?

Response: Using the reviewer’s comment new information was added and now states (pages 8-9, lines 363-373),

It is important to note that in other diseases, manipulation of microbiota has a greater impact than in HCC. It has been demonstrated that animal models of NASH respond to fecal microbiota transplantation (FMT), and early studies have shown that FMT from lean mice donors results in changes in the gut microbiota of obese mice, which are thought to be primarily related to increased microbiota diversity. An ongoing phase I clinical study (NCT02469272) examining small intestinal microbiota transfers from lean to obese subjects has shown improved insulin sensitivity in patients with metabolic syndrome as well as improved insulin sensitivity in those with NASH [37,72,73]. Several factors, including the strength and adaptability of host and donor characteristics, contribute to the variable efficacy of FMT. Research in this area is promising and requires high-quality studies and controlled trials in NASH patients [73,74]”

  1. It is generally recognized that chronic inflammation facilitates cirrhosis or HCC progression, if possible, the immune microenvironment could be further discussed in more detail in a revised version.

Response: Thanks for the reviewer’s comment, new information was added and now states (page 5, lines 185-206)

" The liver microenvironment plays a crucial role in the pathogenesis of HCC, as chronic inflammation, driven by factors inside the liver, facilitates the progression of cirrhosis and HCC. Hepatic stellate cells and tumor macrophages contribute to the induction of fibrosis through the production of extracellular matrix and promote tumor growth, leading to the process of angiogenesis deeply linked to hepatic inflammation [37]. Inflammatory cells such as neutrophils, macrophages and lymphocytes can also produce growth factors and enzymes that support tumor growth and invasion [38].

Neutrophil extracellular traps (NETs) are implicated in the pathogenesis of NASH. Their association with inflammation and globular degeneration highlights their role in the disease. Some studies suggest that the fibrous structure of NETs enhances their bactericidal capacity by sequestering bacteria with a high local concentration of antimicrobial molecules [39]. In addition, IL-1β- and IL-17A-enriched NETs contribute to the hepatic inflammatory process in NASH by providing a vehicle for IL-1β and IL-17A. Furthermore, platelet aggregation in hepatic sinusoids implicates the role of thrombo-inflammation in NASH and may explain the low peripheral blood platelet counts observed in these patients [40].

The two main pro-tumorigenic mechanisms by which immune cells promote HCC include the secretion of cytokines and secretion of cytokines and growth factors that promote proliferation or counteract apoptosis of tumor cells, as well as suppress the anti-tumor function of lymphocytes. In addition, reported results indicate that the NF-κB and JAK-STAT pathways are key inflammatory signaling pathways in the promotion of HCC [41]".

  1. Remove “non” from line 160 page 4 where it says “non-NASH increases the risk of developing HCC compared to patients without NASH”.

Response: Using the reviewer’s comment the “non” was removed from the sentence.

Reviewer 2 Report

It is estimated that a quarter of the world's population suffers from non-alcoholic fatty liver disease. This disease can progress to a more serious form, non-alcoholic steatohepatitis (NASH), a disease with a greater probability of progression to cirrhosis and hepatocellular carcinoma. In this article, we address the link between inflammation, the microbiota, and hepatocarcinoma and review the different in vitro models as well as recent clinical trials addressing liver cancer and the microbiota. The reviewer’s concerns are as follows:

1. It’s suggested to add a table summarizing the clinical trials about liver cancer and the microbiota.

2. The detailed figure legend should be added.

3. It’s suggested to check the grammar and correct the typo errors.

4. Some related reference should be added. For example, line 172.

 It’s suggested to check the grammar and correct the typo errors.

Author Response

Ms. Nadja Spasojevic,

Assistant Editor, MDPI Belgrade,

Thank you for providing us with the opportunity to submit a revised version of our manuscript entitled “From nonalcoholic fatty liver disease to liver cancer: microbiota and inflammation as key players” to the Pathogens journal. We would like to thank the reviewers for their thoughtful comments and suggestions regarding our manuscript. All comments have been considered and incorporated into the revised manuscript as well as the changes according to the Authenticate report. Changes are highlighted in green and blue font for the reviewer's comments with an itemized point-by-point response to the reviewers' comments.

COMMENTS FROM REVIEWER #2

It is estimated that a quarter of the world's population suffers from non-alcoholic fatty liver disease. This disease can progress to a more serious form, non-alcoholic steatohepatitis (NASH), a disease with a greater probability of progression to cirrhosis and hepatocellular carcinoma. In this article, we address the link between inflammation, the microbiota, and hepatocarcinoma and review the different in vitro models as well as recent clinical trials addressing liver cancer and the microbiota. The reviewer’s concerns are as follows:

  1. It’s suggested to add a table summarizing the clinical trials about liver cancer and the microbiota.

Response: Using the reviewer’s comment the table was added on pages 7-8, lines 316-329.

  1. The detailed figure legend should be added. It’s suggested to check the grammar and correct the typo errors.

Response: Using the reviewer's comment, the detailed figure legend was added on page 4, lines 153-156, and grammar and typographical errors were revised.

  1. Some related references should be added. For example, in line 172.

Response: We have added several new references along the text

  1. Comments on the Quality of English Language.  It’s suggested to check the grammar and correct the typo errors.

Response: Apologies for the inconvenience, grammar and typographical errors were revised.

Reviewer 3 Report

I have studied carefully the manuscript entitled "From nonalcoholic faĴy liver disease to liver cancer: microbiota and inflammation as key players" by Rodríguez-Lara A. et al.

The manuscript constitutes an interesting and valuable review on the way intestinal microbiota stimuli and the subsequent inflammatory process might interplay to affect the long way from fat accumulation to tumorigenesis in hepatocytes.

The manuscript is well-prepared and the references are adequate and quite up-to-date. The text is organized in a manner that facilitates the readership, which is expected to be broad, considering the high significance of the topic.

However, before considering publication, the authors are wellcome to discuss the points listed below, in an effort to ameliorate the quality of the manuscript.

Major issues:

1) NETs are implicated in the pathogenesis of NASH, providing a vehicle for IL-1β and IL-17A; this has been documented in histologically-proven NASH in terms of NETs association with inflammation, ballooning degeneration, and fibrosis stage as well as NETs co-localization with IL-1β and IL-17A in a setting of larger than usual platelet aggregates (see: Arelaki S. et al. Neutrophil extracellular traps enriched with IL-1β and IL-17A participate in the hepatic inflammatory process of patients with non-alcoholic steatohepatitis. Virchows Arch. 2022 Sep;481(3):455-465). Therefore, a mention on the potential role of NETs in NASH could further consolidate the potential role of immunothrombosis the inflammatory process.

2) Some additional references could be considered helpful including: i) Effenberger M. et al, A gut bacterial signature in blood and liver tissue characterizes cirrhosis and hepatocellular carcinoma. Hepatol Commun. 2023 Jun 14;7(7):e00182, ii) Kotsiliti E. et al, Intestinal B-cells license metabolic T-cell activation in NASH microbiota/antigen-independently and contribute to fibrosis by IgA-FcR signalling. J Hepatol. 2023 May 9:S0168-8278(23)00325-2, and iii) Li T. et al. Akkermansia muciniphila suppressing nonalcoholic steatohepatitis associated tumorigenesis through CXCR6+ natural killer T cells. Front Immunol. 2022 Dec 1;13:1047570.

Minor issues:

1) English language used needs some polishing.

English language used needs some polishing.

Author Response

Ms. Nadja Spasojevic,

Assistant Editor, MDPI Belgrade,

Thank you for providing us with the opportunity to submit a revised version of our manuscript entitled “From nonalcoholic fatty liver disease to liver cancer: microbiota and inflammation as key players” to the Pathogens journal. We would like to thank the reviewers for their thoughtful comments and suggestions regarding our manuscript. All comments have been considered and incorporated into the revised manuscript as well as the changes according to the Authenticate report. Changes are highlighted in green and blue font for the reviewer's comments with an itemized point-by-point response to the reviewers' comments.

COMMENTS FROM REVIEWER #3

I have studied carefully the manuscript entitled "From nonalcoholic fatty liver disease to liver cancer: microbiota and Inflammation as key players" by Rodríguez-Lara A. et al.

The manuscript constitutes an interesting and valuable review on the way intestinal microbiota stimuli and the subsequent inflammatory process might interplay to affect the long way from fat accumulation to tumorigenesis in hepatocytes.

The manuscript is well-prepared and the references are adequate and quite up-to-date. The text is organized in a manner that facilitates the readership, which is expected to be broad, considering the high significance of the topic.

However, before considering publication, the authors are welcome to discuss the points listed below, in an effort to ameliorate the quality of the manuscript.

Major issues:

  • NETs are implicated in the pathogenesis of NASH, providing a vehicle for IL-1β and IL-17A; this has been documented in histologically-proven NASH in terms of NETs association with inflammation, ballooning degeneration, and fibrosis stage as well as NETs co-localization with IL-1β and IL-17A in a setting of larger than usual platelet aggregates (see: Arelaki S. et al. Neutrophil extracellular traps enriched with IL-1β and IL-17A participate in the hepatic inflammatory process of patients with non-alcoholic steatohepatitis. Virchows Arch. 2022 Sep;481(3):455-465). Therefore, a mention on the potential role of NETs in NASH could further consolidate the potential role of immunothrombosis the inflammatory process.

Response: Thanks for the reviewer’s comment, new information was added and now states (page 5, lines 192-200),

“Neutrophil extracellular traps (NETs) are implicated in the pathogenesis of NASH. Their association with inflammation and globular degeneration highlights their role in the disease. Some studies suggest that the fibrous structure of NETs enhances their bactericidal capacity by sequestering bacteria with a high local concentration of antimicrobial molecules [39]. In addition, IL-1β- and IL-17A-enriched NETs contribute to the hepatic inflammatory process in NASH by providing a vehicle for IL-1β and IL-17A. Furthermore, platelet aggregation in hepatic sinusoids implicates the role of thrombo-inflammation in NASH and may explain the low peripheral blood platelet counts observed in these patients [40]”.

  1. Some additional references could be considered helpful including: i) Effenberger M. et al, A gut bacterial signature in blood and liver tissue characterizes cirrhosis and hepatocellular carcinoma. Hepatol Commun. 2023 Jun 14;7(7):e00182, ii) Kotsiliti E. et al, Intestinal B-cells license metabolic T-cell activation in NASH microbiota/antigen-independently and contribute to fibrosis by IgA-FcR signalling. J Hepatol. 2023 May 9:S0168-8278(23)00325-2, and iii) Li T. et al. Akkermansia muciniphila suppressing nonalcoholic steatohepatitis associated tumorigenesis through CXCR6+ natural killer T cells. Front Immunol. 2022 Dec 1;13:1047570.

Response: Thanks for the reviewer’s comment, the suggested references have been newly added and now states (pages 6-7, lines 288-291)

“Accordingly, Effenberger et al. compared profiles to non-malignant cirrhotic and non-cirrhotic NAFLD patients and found that patients with HCC and cirrhosis exhibited an increased presence of bacterial gene signatures in contrast to NAFLD [65].” Page 5, lines 207-211,  “The role of gastrointestinal B cells in the development of NASH, fibrosis and NASH-induced HCC was also examined. Activated B cells in the gut were elevated in both human and mouse NASH samples. These activated B cells were shown to contribute to NASH development independently of antigen specificity and gut microbiota by promoting metabolic activation of T cells [42]. And page 6, lines 273-274, “They observed lower levels of Akkermansia in both, NASH-HCC patients and mice, with [62].”

Minor issues: English language used needs some polishing.

Response: Apologies for the inconvenience, grammar and typographical errors were revised.
